# Impact of Power and Time in Hepatic Microwave Ablation: Effect of Different Energy Delivery Schemes

**DOI:** 10.3390/s24237706

**Published:** 2024-12-02

**Authors:** Macarena Trujillo, Mahtab Ebad Najafabadi, Antonio Romero, Punit Prakash, Francois H. Cornelis

**Affiliations:** 1BioMIT, Electronic Engineering Department, Universitat Politècnica de València, 46022 Valencia, Spain; 2Department of Biomedical Engineering, George Washington University, Washington, DC 20052, USA; mahtabe@gwu.edu (M.E.N.); punit.prakash@gwu.edu (P.P.); 3Department of Applied Mathematics, Universitat Politècnica de València, 46022 Valencia, Spain; aromtar@mat.upv.es; 4Department of Radiology, Memorial Sloan Kettering Cancer Center, New York, NY 10065, USA; cornelif@mskcc.org

**Keywords:** microwave ablation, theoretical model, power delivery scheme

## Abstract

Microwave ablation often involves the use of continuous energy-delivery protocols with a fixed power and time. To achieve larger ablation zones, a range of protocols and power levels have been studied in experimental studies. The objective of the present study was to develop and experimentally evaluate the performance of a coupled computational electromagnetic–bioheat transfer model of 2.45 GHz microwave ablation under a variety of continuous and pulsed power delivery schemes. The main aim was to obtain an in-depth knowledge of the influence of energy delivery settings on ablation zone profiles and thermal damage in the peri-ablation zone. In addition to the theoretical model, we evaluated the power delivery schemes using ex vivo experiments and compared them to previously published data from in vivo experiments. The results showed slight differences in terms of the ablation zone size for different power delivery schemes under ex vivo conditions, with the applied energy level being the most important factor that determines ablation zone size; however, under in vivo conditions, applying a high-power pulse prior to and following a longer constant power application (BOOKEND 95 W protocol) presented the most favorable ablation zones. Moreover, the modeling and experimental studies identified threshold applied power and ablation times beyond which increases did not yield substantive increases in ablation zone extents.

## 1. Introduction

Image-guided microwave ablation is an established modality for the minimally invasive treatment of localized primary liver tumors and liver metastases [1]. The objective of an ablation treatment is to encompass the targeted tumor and a ~10 mm circumferential margin of adjacent normal tissue with an ablative thermal dose [2]. Analyses of clinical outcomes following ablation procedures have demonstrated the significance of establishing adequate treatment margins for achieving local control [3]. A challenge for clinical operators during ablation procedures, however, is how to select treatment delivery parameters that are likely to yield an adequate treatment margin, such as applied power, ablation duration, and the number of ablation applicators (and their position relative to the targeted tumor). This challenge is exacerbated by the non-linear relationship between applied energy level and size and shape of the ablation zone, and the influence of tissue physical properties, which vary considerably as a function of tumor and normal tissue state and change dynamically as a function of temperature and tissue hydration level during an ablation procedure [4,5,6,7,8,9]. With increasing interest in combination therapies involving ablation that harness the hyperthermic temperatures at the ablation zone periphery, understanding how treatment delivery factors influence induced temperature profiles beyond the ablation zone is also of interest.

The bulk of microwave ablation studies have employed a fixed applied power level over the course of the ablation procedure. To achieve larger ablation zones, the number of applicators, applied power level, and/or ablation duration are increased. As tissue heats and desiccates during the ablation process, tissue dielectric properties drop considerably at temperatures exceeding ~80 °C, impacting the transfer of microwave power from the applicator to tissue, the profile of the electric field within the tissue, and subsequently, the rate of heating [4,10]. Transient modulation of the applied power has been investigated as a means for limiting peak tissue temperature and maintaining efficient transfer of power during the ablation procedure. Power modulation schemes that have been investigated include pulsing, ramped power, and “book-ending”, a constant power period with high power bursts [11]. The bulk of studies have employed an experimental approach, evaluating the influence of the power delivery algorithm on the extent of the ablation zone and temperature profiles in ex vivo and in vivo tissue. While experimental approaches, particularly those in large animals in vivo, provide a high level of evidence, they do not consider the role of several factors that may considerably influence bioheat transfer dynamics during ablation, notably the contrast in tissue physical properties between tumor and surrounding normal tissue. Computational models provide an approach to complement findings from experimental studies that enable consideration of such factors, provide insight into the biophysical factors influencing ablation outcomes [12,13], and enable consideration of a large range of energy delivery settings, from which a subset can be selected for experimental evaluation [14].

The objective of the present study was to develop and experimentally evaluate the performance of a coupled computational electromagnetic–bioheat transfer model of 2.45 GHz microwave ablation under a variety of continuous and pulsed power delivery schemes. Model predicted extents of ablation zone were qualitatively assessed against experimental data from in vivo experimental studies reported in the literature, and quantitatively assessed against data from ex vivo experimental studies presented herein. The presented model is intended to be used to guide the design of in vivo experimental studies investigating the role of microwave energy delivery algorithms on the extent of the ablation zone and the extent and characteristics of thermal damage in the peri-ablation zone.

## 2. Materials and Methods

### 2.1. Computational Model

#### 2.1.1. Geometry

The computational model mimicked microwave ablation with a water-cooled 2.45 GHz applicator. Figure 1A illustrates the model geometry, consisting of the ablation applicator inserted within a homogeneous block of normal liver tissue. We considered a two-dimensional model assuming the axial symmetry of the ablation applicator. The tissue region was modeled as a cylinder of 50 mm radius and 100 mm height. These dimensions, obtained from a convergence test (see Section 2.1.4), afforded approximation of the applicator as being inserted within an infinitely large tissue domain, such that the distance between the applicator and tissue boundaries did not impact simulation results. The applicator incorporated a coaxial monopole antenna, with a 7.5 mm radiating tip, enclosed within concentric thin-walled plastic tubing to enable closed-loop water circulation, as previously described in [15]. Details of the antenna and applicator geometry are shown in Figure 1B,C.

#### 2.1.2. Governing Equations and Conditions

The model was implemented as a coupled electromagnetic-thermal problem, which was solved numerically using Comsol Multiphysics (Comsol, Burlington, MA, USA). The governing equation for the thermal problem was the Bioheat equation [16], modified by the enthalpy method to include water vaporization [17]:(1)$$\frac{\partial (\rho h)}{\partial t}=\nabla \cdot \left(k\nabla T\right)+Q_{b}+Q_{m}+Q_{e}$$
where ρ (kg/m^3^) is tissue density, h (J/kg·K) enthalpy, k (W/m^2^·K) thermal conductivity, T (K) temperature, $Q_{b}$ (W/m^3^) heat loss by blood perfusion, $Q_{m}$ (W/m^3^) the metabolic heat generation (negligible for MWA, and thus not included in the model), and $Q_{e}$ (W/m^3^) refers to the heat source produced by MW power.

For biological tissues, enthalpy is a piecewise function of temperature [17]:(2)$$\frac{\partial (\rho h)}{\partial t}=\frac{\partial T}{\partial t}\cdot \left\{\frac{\rho_{l}c_{l}+{\;\rho }_{g}c_{g}}{2}+\begin{array}{c} \rho_{l}c_{l}\;\;\;\;\;\;\;\;\;\;\;\;\;\;\;\;\;\;\;\;\;\;\;\;\;\;\;\;\;0<T\leq 95\;℃\\ h_{fg}C\;(\frac{1}{105-95})\;\;\;\;\;\;\;\;\;\;\;95<T\leq 105\;℃\\ {\;\rho }_{g}c_{g}\;\;\;\;\;\;\;\;\;\;\;\;\;\;\;\;\;\;\;\;\;\;\;\;\;\;\;\;\;\;\;T>105\;℃ \end{array}\right.$$
where $\rho_{i}$ and $c_{i}$ are the density and specific heat of tissue at temperatures below 95 °C (i=l) and above 105 °C (i=g), $h_{fg}$ (J/m^3^) is the product of water’s latent heat of vaporization and the water density at 100 °C (1000 kg/m^3^), and C(%) is the water content inside the tissue (73%).

The blood perfusion term was only considered for in vivo conditions (see Section 2.1.5). This term was obtained from the expression:(3)$$Q_{b}=\beta \rho_{b}c_{b}\omega_{b}\left(T_{b}-T\right)$$
where $\rho_{b}$ (kg/m^3^) is the blood density, $c_{b}$ (J/kg·K) is the blood-specific heat, $T_{b}\;$(K) the blood temperature (310 K or 37 °C) and $\omega_{b}$ (0.019 s^−1^) the blood perfusion coefficient. The thermal damage in tissue results in cessation of perfusion, then parameter β takes the values of 0 or 1 depending on the thermal damage (β=0 if Ω≥ 4.6, β=1 if Ω<4.6). The parameter Ω was assessed by the Arrhenius thermal damage model [18]:(4)$$\Omega \left(T\right)=\int_{0}^{t}Ae^{-\frac{∆E}{RT}}\;\;ds$$
where R is the universal gas constant, A (7.39·10^39^ s^−1^) is a frequency factor, and ∆E (2.577·10^5^ J/mol) is the activation energy for the irreversible damage reaction.

The time-averaged absorbed electromagnetic power density, $Q_{e},$ was computed from the electromagnetic field distribution in tissue E (V/m), and is given by
(5)$$Q_{e}=\frac{1}{2}\sigma {\left|E\right|}^{2}$$
where σ (S/m) was the (effective) conductivity. The time-harmonic Helmholtz electromagnetic equation was solved to determine E
(6)$$\nabla \times \mu_{r}^{-1}\left(\nabla \times E\right)-k_{0}^{2}\left(\varepsilon_{r}-\frac{j\;\sigma }{\omega \varepsilon_{0}}\right)E=0$$
$\mu_{r}$ being the relative permeability ratio, which is the unity of all the materials in this model, $k_{0}$ is the propagation constant in free space, $\varepsilon_{r}$ is relative permittivity, $\varepsilon_{0}\;$(F/m) is the permittivity of free space, and ω (rad/s) is the angular frequency.

The thermal boundary conditions were initial tissue temperature and temperature at all the outer boundaries of $T_{a}\;$($T_{a}=$37 °C for in vivo conditions and $T_{a}=$24 °C for ex vivo conditions, see Section 2.1.5) and null heat flux on the symmetry axis. The cooling effect of the liquid circulating inside the antenna was modeled using the Newton’s cooling law:(7)$$n\cdot \left(k\nabla T\right)=h\left(T_{r}-T\right)$$

A thermal convection coefficient, *h*, with a value of 3184 W/K∙m^2^ and a coolant temperature of 8 °C. The value of *h* was calculated for a flow rate of 80 mL/min through an area estimated as half of the internal area of the water circuit (see Figure 1C).

The electromagnetic boundary conditions were: null initial electric field, null flux in the symmetry axis, and a first-order electromagnetic scattering boundary condition was applied at the outer boundaries:(8)$$n\times \left(\nabla \times E\right)-jkn\times \left(E\times n\right)=0$$

A perfect electric conductor boundary condition was imposed at the surface of metallic regions of the antenna. Finally, the input power (see Section 2.1.5 for values and modes) was set as a coaxial port boundary condition at the top of the antenna. This coupled electromagnetic-heat transfer model was previously benchmarked against experimentally measured transient temperature profiles using volumetric MR thermometry [15].

#### 2.1.3. Characteristics of Materials

Table 1 shows the constant parameter values used in the model [17,19,20]. In the case of the thermal conductivity k, the effective conductivity σ and the relative permittivity $\varepsilon_{r}$, we used thermal-dependent functions. We adopted the functions used in [20] for the electromagnetic parameters:(9)$$\varepsilon_{r}(T)=44.3\left(1-\frac{1}{1+e^{5.223-0.524T}}\right)$$
(10)$$\sigma \left(T\right)=1.8\left(1-\frac{1}{1+e^{6.583-0.598T}}\right)$$

And the function used in [15] for the thermal conductivity:(11)$$k\left(T\right)=\left\{\begin{array}{c} 0.5+0.003\;\left(T-25\right)\;\;\;\;\;\;\;\;\;\;T<100\;℃\\ 0.5+0.003\;\left(100-25\right)\;\;\;\;\;\;T>100\;℃ \end{array}\right.$$

#### 2.1.4. Solver 

We employed a heterogeneous mesh with triangular elements, with maximum and minimum element edge lengths of 6.7 mm and 0.03 mm, respectively. Additional mesh refinement was defined on the following boundaries and domains: 0.01 mm maximum element length was specified at the port boundary; 0.1 mm maximum element length was specified in all domains within the applicator; and 3.5 mm maximum element length within the tissue domain. The transient solver employed an implicit time-stepping scheme with the backward differentiation formula (BDF), considering a maximum time step of 0.1 s.

The model was verified by convergence tests in terms of domain outer dimensions. The reference parameter for the convergence tests was the maximum temperature ($T_{max}$) reached after 3, 5, and 10 min at three points 10, 20, and 30 mm from the antenna surface. The domain’s outer dimensions were those of the reference [15]. These dimensions were changed by +1 mm, and we assessed the changes in $T_{max}$. The former dimensions were used in the model for $T_{max}$ differences of less than 0.5% between the two subsequent simulations.

#### 2.1.5. Computational Modeling Phases

Phase 1 was used to assess the effect of different power levels and times on axial (L) and transversal (D) diameters of the ablation zone. Both effects were examined under ex vivo and in vivo conditions. Powers ranging from 10 to 100 W (in 10 W steps) were used, and the D and L values were checked after 5 and 10 min of continuous power application. Special attention was paid to three representative powers: 50, 75, and 100 W. We performed ablation experiments in ex vivo tissue to compare to modeling results of these three power levels (see Section 2.2 Experimental model).

Phase 2 was used to explore the effect of using different application power-time protocols on L and D. In this sense, the protocols used were the same as those presented in [11] under in vivo conditions to compare with this reference results qualitatively and under ex vivo conditions to quantitatively compare with experimental ex vivo results (see Section 2.2, Experimental model). First, four protocols with the same level of energy (39 kJ) were considered (see Figure 2, in which the X-axis represents time in seconds and the Y-axis represents power in Watts):65 W 10 MIN: A continuous application of 65 W for 10 min.RAMPED: A step-wise increase in which power starts with a low value of 25 W and ends with 65 W: 25 W-30 s, 35 W-30 s, 45 W-30 s, 55 W-30 s, 65 W-8 min, and 46 s.LOW POWER: A continuous application of 40 W for 16 min and 15 s.95 W PULSED: A periodic application of 95 W pulses with 31–32 cooling pauses.

Two higher energy protocols were also used (see Figure 3, in which the X-axis represents time in seconds and the Y-axis represents power in Watts):BOOKEND 95 W: A continuous 65 W for 8 min is applied, preceded and followed by 95 W for 1 min. The total applied energy of this protocol is 42.6 kJ.65 W 15 MIN: A continuous application of 65 W for 15 min. The total applied energy of this protocol is 58.5 kJ.

The mentioned power in Phases 1 and 2 refers to the power setting at the generator. In phase 1 (in which no comparison with the [11] was made), the power applied at the port boundary condition of the theoretical model was 80% of the specified power level, an estimation of the actual power transferred to the antenna used in the ex vivo experiments, after considering attenuation within inter-connecting cables. In phase 2, the actual power transferred to the antenna was 65% of the power setting at the generator, as reported by [11].

#### 2.1.6. Outcomes

We employed the Ω= 4.6 contour (Equation (4)) to estimate the extent coagulation zone, which corresponds to 99% of cell death. Axial (L) and transversal (D) diameters were the parameters used to measure the extent of the coagulation zone.

### 2.2. Experiments in Ex Vivo Bovine Liver Tissue

Microwave ablation experiments were performed in ex vivo bovine liver tissue for assessment of computational model performance. We used fresh bovine liver within 36 h of excision from the animal; livers were transported from a local slaughterhouse to the lab in sealed plastic bags, which were placed on ice in a cooler. Prior to starting ablation experiments, the liver tissue was sectioned into approximately ~150 cm^3^ blocks, placed in sealed plastic bags, and placed in a temperature-controlled bath to warm up to ambient temperature (i.e., approximately 24 °C).

Custom water-cooled ablation applicators were prepared for these experiments, as previously described [15]. Microwave power was supplied by a 2.45 GHz solid-state microwave generator (Leanfa Srl, Bari, Italy) controlled by custom software implemented with Matlab to keep track of forward and reflected power. Chilled water (~7 °C) was circulated through the applicator using a peristaltic pump (Masterflex, Cole-Parmer, Vernon Hills, IL, USA). To compare to the modeling results of Phase 1, we conducted experiments of 50, 75, and 100 W using 5 and 10 min. In the case of Phase 2, all the protocols considered were conducted. Experiments for each setting were repeated a minimum of *n* = 3 times. After the ablation, the liver tissue was transversally sectioned to measure the transversal and axial diameters of the ablation zone using a ruler with an accuracy of ±1 mm. All the measurements were made over the external “pink zone” (see example annotation in the top left image in Figure 4 where diameters L and D are marked). 

## 3. Results

Figure 4 depicts example ablation zones for each of the protocols considered in the ex vivo experiments.

In Phase 1, we were interested in assessing the effect of different power levels and times on axial and transversal diameters of the coagulation zone (L and D). Figure 5 and Figure 6 present the computational model predictions of ablation zone extents (D and L) for applied input powers of 50, 75, and 100 W. We found that the axial diameter L generally increased with increasing applied power, although the influence of applied power on D diminished for power levels above 40 W. Under ex vivo conditions, differences became negligible after 500 s between 75 and 100 W. Despite that, in the case of L value, we observed more substantial differences between the different power levels; under in vivo conditions, differences between 75 W and 100 W were also negligible after 480 s.

Figure 6 shows D and L values as a function of the power level in a range from 10 to 100 W and for 5 and 10 min of MWA under ex vivo and in vivo conditions. We observed significant differences between the D value for 5 and 10 min under both ex vivo and in vivo conditions; however, these differences were smaller in the case of L value under ex vivo conditions and for low power values (P < 50 W), and almost negligible under in vivo conditions. For transversal diameter D, data from ex vivo simulations show that the value tends to saturate with an increase in power for both times (5 and 10 min): the D value does not increase from approximately 75 W; however, the L value continues increasing with the power after 100 W, also under in vivo and ex vivo conditions and for both considered times.

Table 2 shows the comparison between ex vivo computational and experimental results for D and L diameters obtained for 50, 75, and 100 W after 5 and 10 min of MWA. The error refers to the relative difference between the computational and experimental measures, using the mean value for the experimental measures. Higher differences between computational and experimental measures were found for L than for D, being 8.3% of the mean error for D and 9.9% for L, and the maximum error found was 15%. Positive error values in all cases indicate that the computational values obtained are always higher than the corresponding experimental values.

The aim of Phase 2 of the study was to assess the size of the ablation zone using the same protocols as those presented in [11], under in vivo conditions (to compare with this reference) and under ex vivo conditions (to compare with our experimental results). In this sense, Table 3 shows ex vivo computational and experimental values for D and L diameters obtained using the protocols of Phase 2. Again, the error means the relative difference between theoretical and experimental measures, using the mean value for the experimental measures. As in Phase 1, all the errors were positive since the values computed from the computational model were higher than the experimental values. The mean error for D and L were 6.4% and 3%, respectively, and the maximum error found was 13%.

From Table 3, we can assess the effect of protocols and levels of energy under ex vivo conditions. According to the computational modeling results, the difference between all the cases with 39 kJ of energy (65 W 10 MIN, RAMPED, LOW POWER, and 95 W PULSED) was 3 and 6% for D and L, respectively, and these percentages were 6 and 10% based on the mean of the experimental results. In this last case, considering the standard deviation, we could find that there was a range of 2 mm amplitude for D and L in which all the values coincided. In the BOOKEND 95 W protocol, with an energy level of 3.6 kJ higher (9% more energy than the previous cases), the theoretical and experimental values of D and L are higher; however, these differences with any of the previous cases are under 7% (except for L value measured experimentally). From the standard deviation, the values of D and L for the BOOKEND 95 W protocol also matched with the values range of the protocols of 39 kJ. For the highest energy level protocol (65 W 15 MIN, 58.5 kJ), we obtained the highest values of D and L in experiments and modeling, obtaining in most of the cases differences between 4–5 cm for D and L, which represents an increase of ≈9–10%.

Moreover, to qualitatively assess the in vivo results obtained in Phase 2, we compared the results obtained using the computational model and those obtained experimentally in [11]. The aim of this comparison was only from a qualitative point of view since the conditions of this experiment were not identical to those for the computational model (e.g., different applicator, cooling mechanism). Figure 7 shows the results of this comparison in which the same trend was observed in theoretical and experimental results: (1) Almost all the protocols with the same level of energy (65 W 10 MIN, RAMPED, and 95 W PULSED) produced similar values of D and L values, with the exception of the LOW POWER protocol which produces smaller values, (2) the BOOKEND 95 W, with a moderate increase in energy with respect to the previous protocols (42.6 kJ vs. 39 kJ), produced higher values of D and L values, (3) the 65 W 15 MIN protocol, in which the increase in energy with respect to BOOKEND 95 W was remarkable, did not produce a noticeable increase in the D and L values respect to that protocol. Despite the trend being the same, we observed that in [11], the differences in the value of D and L between the protocols of BOOKEND 95 W and 65 W 15 MIN and the rest of the protocols are much more noticeable. Moreover, in the case of D, the experimental results show a higher value in the case of BOOKEND 95 W than for 65 W 15 MIN, which was not observed in the theoretical results.

## 4. Discussion

In the present study, our overall goal was to employ computational models to study how the dimensions of the ablation zone varied in relation to the applied power, time, and total energy and to benchmark model performance against experimental data. A benchmarked model would support the application of the model towards identifying energy-delivery protocols most suitable for achieving desired extents of the ablation and peri-ablation zone in future studies, with application to an investigation of ablation employed in combination with other therapeutic modalities.

The modeling results obtained in Phase 1 were aimed to identify thresholds of power and time in which the growth of the coagulation zone is limited and to identify the conditions of constant power protocols that more efficiently ablate tissue. This aligned with one of the main aims of the works in the MWA field: the research about how to obtain near-spherical ablation zones large enough to ablate tumors and a 0.5–1 cm margin of surrounding normal tissue [13,21,22].

With this objective in mind, as we can see in Figure 5 and Figure 6, high powers or longer times, although they produce more elongated coagulation zones, are needed to ablate ~2–3 cm diameter targets. This is a known characteristic of the MWA, sometimes referred to as the “comet effect” [23,24]. This is produced by an unequal growth of D and L: the growth rate of D is lower than that of L. This growth rate is not linear in any case (L and D) since a parabolic tendency is observed; however, the radius of curvature is always higher for L than for D. Under ex vivo conditions, D reaches a limiting value after 600 s for 100 W, which is not observed for L. From Figure 5 and Figure 6, we can observe that after 600 s, increasing applied power from 75 W to 100 W does not yield increased transversal ablation diameter D. In summary, this suggests a threshold applied power level above which there is no benefit observed in terms of D: powers higher than 75 W do not produce wider coagulation zones than using 100 W after 600 s.

One of the main reasons for this unequal growth of L and D is the impedance mismatch of the antenna to the target tissue, which increases with power and, consequently, more rapid changes in tissue dielectric properties, resulting in a higher reflection coefficient [23,25]. To such an extent that, as can be seen, the application of higher power (as in the case of 100 W vs. 75 W) may not result in an increase in the size of the coagulation zone due to higher antenna reflection losses and reduced power transferred to the tissue. The conditions of the in vivo models, which consider the sink effect, show that this effect is more marked and that there are hardly any differences in the dimensions obtained with 75 and 100 W, and even that with 75 W higher values of diameter D are obtained from 450 s onwards. In the in vivo setting, the blood perfusion heat sink poses a further barrier to the growth of the ablation zone by thermal conduction.

It is important to remark that the time and power thresholds vary with the antennas since, depending on their characteristics, the losses or the reflection coefficient are different.

The comparison between the experimental and theoretical results of Phase 1 (Table 2) shows that both are in good alignment with the mean of errors under 10%. A similar result was also obtained in Phase 2 (Table 3). This similarity between experimental and theoretical results supports the further use of the theoretical model for comparative assessment of various power delivery algorithms. As we can observe, although differences are small, they are always positive, which implies that higher values are obtained with the theoretical model. This effect is also found in other works [26,27], and one of the main reasons that can explain it is that the model does not consider tissue shrinkage, which does occur during ablation studies and which is estimated to account for between 10 and 15% of the value of the diameters obtained in the theoretical model.

Results in Phase 2 (Table 3 and Figure 7) allowed comparison of the size of the ablation zone for varying applied power protocols. Under ex vivo conditions, there were no substantial differences between all the protocols with the same level of energy (65 W 10 MIN, RAMPED, LOW POWER, and 95 W PULSED), and almost negligible differences are noticed with the BOOKEND 95 W protocol with a slightly higher level of energy. These protocols can be considered equivalent in terms of sphericity of coagulation zone diameters, due to the similarity of their results. Generally, higher values for D and L are obtained when using protocols with higher energy levels, although relatively large increases in applied energy are required to achieve relatively modest increases in D and L. From the point of view of the comparison of experimental and theoretical results, the same remarks as in Phase 1 can be made: the theoretical results are always higher than the experimental, and they are validated with very similar experimental results.

In Phase 2, we aimed to assess the model-predicted ablation zones qualitatively as a function of applied energy levels were analogous to those experimentally obtained by Hui et al. [11] while noting the ablation antenna considered in our study (in modeling and ex vivo experiments) was different. As Table 3 depicts, similar trends to that of the study of Hui et al. were observed, however, the theoretical results show that more homogeneous results are obtained for protocols with the same level of energy. Further, the size of the diameters D and L are related to the level of energy of the protocol and have little dependence on the type of protocol. While the in vivo experimental results of [11] showed that BOOKEND 95 W and 65 W 15 MIN had great differences in terms of D and L with the rest of the protocols, this trend was not observed in our modeling results. Both our modeling results and the experimental results from Hui et al. confirm that LOW POWER is the protocol that produces the smaller coagulation zone under in vivo conditions. Ref. [28] reported that no differences in the ablation zone size were observed between continuous and pulsed protocols under ex vivo and in vivo conditions. In that work, the antenna was different from the one used in the present work, but the results agree qualitatively with those obtained for the 60 W 10 MIN and 95 W PULSED protocols. Moreover, in [29], authors found differences in the size of the carbonization zone during MWA using different aperiodic pulsed protocols; however, no remarkable differences were found in the ablation zone.

These data support the use of the computational model for guiding the selection of energy delivery protocols for in vivo experimental studies aiming to achieve ablation zones within a certain size range. With increasing interest in combination treatments involving thermal ablation, where combination effects may also depend on temperatures in the peri-ablation zone, the presented computational models may also find application in quantitively analyzing thermal profiles in the per-ablation region. Although the results were encouraging, there are still general issues in computational models for MWA that need to be reviewed. A more clinically oriented computer model, which would include more realistic characteristics for some phenomena and properties such as shrinkage, vaporization, or the effective conductivity and relative permittivity modeling, would allow obtaining closer results with experimental models. The overestimation of the coagulation zone and the more marked comet effect of computational models could be corrected using the change in water content or tissue composition during MWA. Finally, further efforts to compare model predictions against clinical results (e.g., ablation zones observed on post-procedural imaging) are warranted.

## 5. Conclusions

The developed computational model for the 2.45 GHz microwave ablation model predicts the coagulation zone obtained experimentally under a variety of continuous and pulsed power delivery schemes. Modeling and experimental findings showed that the level of applied energy is the primary driver of differences in the size of the ablation zone, although in vivo conditions are more sensitive to the type of protocol than ex vivo. The existence of thresholds under ex vivo and in vivo conditions of continuous MWA shows that long times and high-power applications do not always produce better results in terms of the coagulation size and shape. The presented computational model can be used to comparatively assess the use of different microwave energy delivery algorithms and estimate the thermal damage produced in the peri-ablational zone.

## Figures and Tables

**Figure 1 sensors-24-07706-f001:**
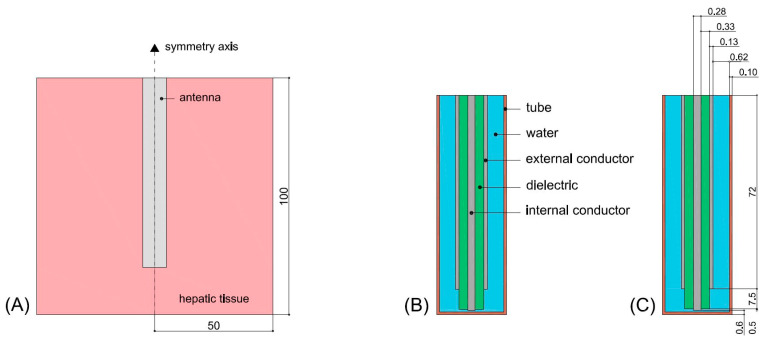
(**A**) Scheme of the geometry used in the computational model. (**B**) Antenna scheme. (**C**) antenna’s dimensions. All dimensions are in mm, and the schemes are not drawn to scale.

**Figure 2 sensors-24-07706-f002:**
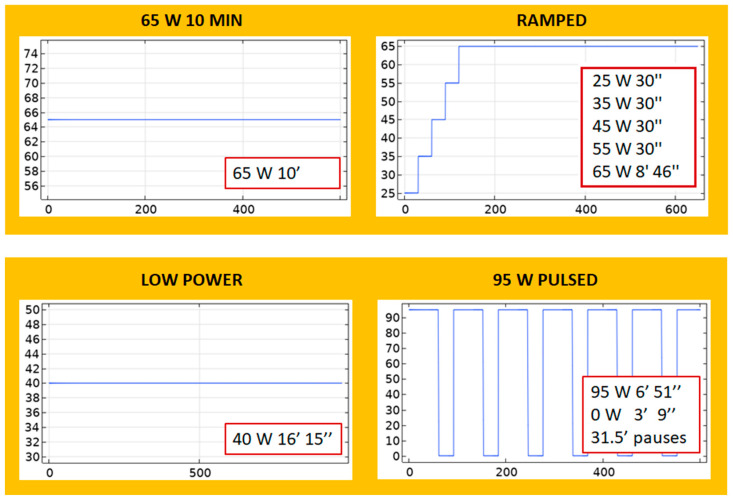
Different protocols used in Phase 2 of the theoretical model with the same level of energy (39 kJ). The protocols are extracted from [11]. The X-axis represents time (s), and the Y-axis represents power (W). The legends specify the on and off times in minutes (′) and seconds (″).

**Figure 3 sensors-24-07706-f003:**
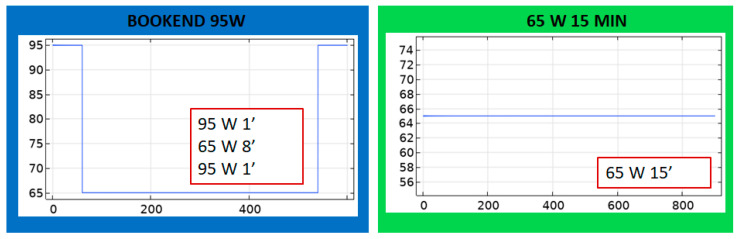
Different protocols used in Phase 2 of the theoretical model with different levels of energy: 42.6 kJ in BOOKEND and 58.5 kJ in 65 W 15 MIN. The protocols are extracted from [11]. The X-axis represents time (s), and the Y-axis represents power (W). The legends specify the on and off times in minutes (′).

**Figure 4 sensors-24-07706-f004:**
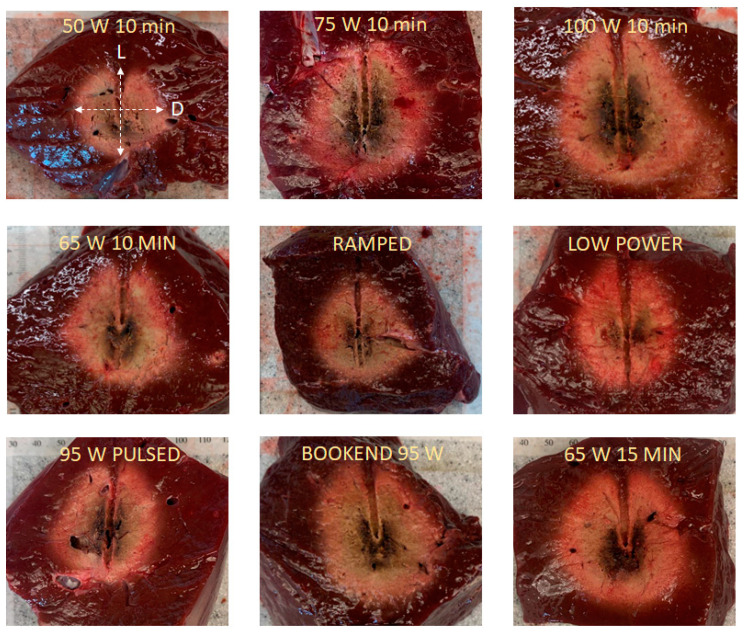
Images of example coagulation zones obtained for the protocols evaluated with ex vivo experiments.

**Figure 5 sensors-24-07706-f005:**
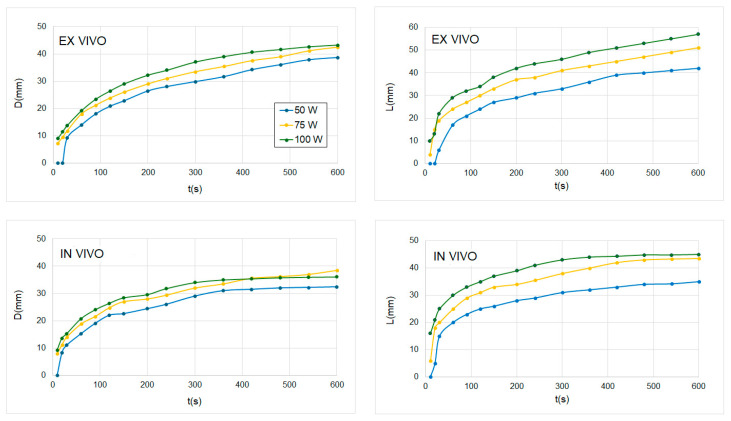
**Left**: Ablation zone transversal diameter (D) evolution for ex vivo (**up**) and in vivo (**down**) conditions for three power levels: 50, 75, and 100 W. **Right**: Ablation zone axial diameter (L) evolution for ex vivo (**up**) and in vivo (**down**) conditions and for three power levels: 50, 75, and 100 W.

**Figure 6 sensors-24-07706-f006:**
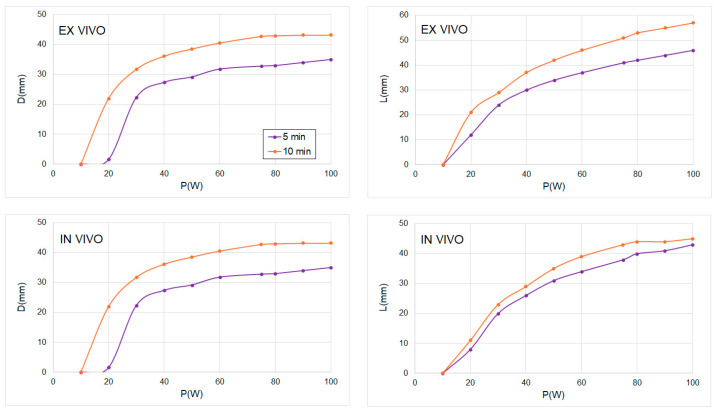
**Left**: Ablation zone transversal diameter (D) as a function of power (P) for ex vivo (**up**) and in vivo (**down**) conditions and for two times: 5 and 10 min. **Right**: Ablation zone axial diameter (L) as a function of power (P) for ex vivo (**up**) and in vivo (**down**) conditions and for two times: 5 and 10 min.

**Figure 7 sensors-24-07706-f007:**
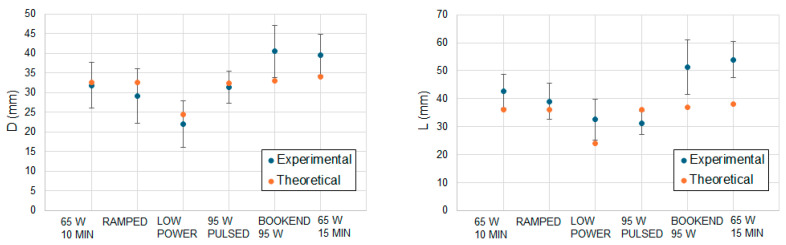
In vivo theoretical and experimental values obtained for D and L diameters in cases of Phase 2 of the study. In vivo results were extracted from [11].

**Table 1 sensors-24-07706-t001:** Characteristics of materials used in the theoretical model [17,19,20].

Material	*σ* (S/m)	$\varepsilon_{r}$	k (W/m·K)	*ρ* (kg/m^3^)	c (J/kg·K)
Liver	1.8 ^(a)^	44.3 ^(a)^	0.5 ^(b)^	1050 ^(c)^	3400 ^(c)^
				370 ^(d)^	2156 ^(d)^
Copper	5.998 × 10^7^	1			
Teflon	10^−4^	2.1			
Tube	0	3.1			
Water	0.9	78			
Blood				1000	4148

*σ*, (effective) conductivity; $\varepsilon_{r}$ relative permittivity; k, thermal conductivity; *ρ*, density; c, specific heat. ^(a)^ Measured at 37 °C, ^(b)^ Measured at 25 °C, ^(c)^ for temperatures between 37 to 99 °C, ^(d)^ for temperatures higher than 100 °C.

**Table 2 sensors-24-07706-t002:** Ex vivo theoretical and experimental values obtained for ablation zone D and L diameters and error between these measures for cases considered in Phase 1 of the study.

		D (mm)			L (mm)		
		Theoretical	Experimental	Error	Theoretical	Experimental	Error
50 W	5 min (n = 3)	29.8	26.2 ± 1.0	12%	32.1	27.6 ± 4.3	15%
	10 min (n = 3)	38.6	35.0 ± 1.0	9%	42.2	36.4 ± 2.6	13%
75 W	5 min (n = 3)	33.5	30.3 ± 2.1	10%	41.1	36.3 ± 2.5	11%
	10 min (n = 4)	42.5	38.2 ± 1.7	10%	51.0	47.8 ± 2.4	6%
100 W	5 min (n = 3)	37.0	34.1 ± 5.1	8%	46.1	42.3 ± 2.3	8%
	10 min (n = 5)	43.2	41.5 ± 2.8	4%	57.0	51.9 ± 6.7	9%

**Table 3 sensors-24-07706-t003:** Ex vivo theoretical and experimental values obtained for ablation zone D and L diameters and error between these measures for cases considered in Phase 1 of the study.

	D (mm)			L (mm)		
	Theoretical	Experimental	Error	Theoretical	Experimental	Error
65 W 10 MIN (n = 5)	39.0	34.9 ± 2.9	11%	45.1	39.3 ± 2.4	13%
RAMPED (n = 4)	39.2	37.4 ± 3.4	5%	42.1	41.9 ± 2.9	0%
LOW POWER (n = 2)	39.1	35.5 ± 0.7	9%	42.0	37.4 ± 3.5	11%
95 W PULSED (n = 4)	38.0	35.3 ± 2.4	7%	45.0	40.7 ± 6.1	10%
BOOKEND 95 W (n = 5)	39.8	38.5 ± 1.7	3%	45.2	44.9 ± 3.1	0%
65 W 15 MIN (n = 2)	44.0	41.0 ± 0.0	7%	49.1	43.5 ± 2.1	11%

## Data Availability

Data are contained within the article.

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
