# Peer review of "Impact of Power and Time in Hepatic Microwave Ablation: Effect of Different Energy Delivery Schemes"

_sensors, 2024, doi:10.3390/s24237706_

Round 1
Reviewer 1 Report
Comments and Suggestions for Authors
Please see the attached field.

Author Response
The authors present an in-depth knowledge of the influence of energy delivery settings on ablation zone profiles and thermal damage in the peri-ablational zone. It is eye-catching and exciting to a researcher in this field. However, moderate corrections and explanations could be done previous to publish the paper in Sensors. Some of my comments and questions on this manuscript are as follows:
COMMENT #1: “Abstract: (1) It is incorrect to say that the aim of microwave ablation is ‘large ablation zones’. The aim of microwave ablation is to achieve complete necrosis, encompassing both the area and depth of the targeted tissue, while maintaining procedural safety and efficacy. (2) Please ensure that the abstract includes key quantitative outcomes and relevant data”.
Response: We agree with the reviewer that to say that the aim of MWA is to achieve a larger ablation zone is not precise, so we have changed the sentence related to this. With respect to the inclusion of key and relevant data in the introduction, we believe that the abstract provides information about the main question of the study: How is the ablation zone size affected by different energy delivery schemes? In this respect, we highlight in the abstract the most interesting results: 1) slight differences under ex vivo conditions, 2) the energy level is the factor which has the most influence on ablation zone extents, 3) the confirmation using theoretical modeling that the BOOKEND 95 W is the protocol that presents the most favorable results under in vivo conditions, 4) and the confirmation that there exist thresholds of applied power (applying energy above this threshold does not imply larger ablation zones). However, we have added some quantitative data related with the main findings of the study to the abstract.
Change in manuscript: We have modified the abstract accordingly.
COMMENT #2: “Keywords: Please enhance the abstract by incorporating 1-2 additional terms that are both specific and precise to describe the thermal effects or other relevant outcomes of the study.”
Change in manuscript: We have added the words: pulsed and continuous protocols.
COMMENT #3: “Page 3: The abbreviations 'MWA' and 'MW' should be fully expanded upon their initial mention to ensure clarity and comprehension.”
Change in manuscript: We have introduced the abbreviation MWA at the beginning of the introduction, and MW is fully expanded in page 3 where it is first mentioned.
COMMENT #4: “In section 2.1.5: What is the source of this parameter data? Please provide the source citation for this parameter.”
Response: We understand that the reviewer is referring to the parameter of section 2.1.6 (rather than 2.1.5). The reference for this parameter can be found for example in:
Paruch, M. (2019). Mathematical modeling of breast tumor destruction using fast heating during radiofrequency ablation. Materials, 13(1), 136.
Change in manuscript: We have added this reference in section 2.1.6 and in the list of references. The number of references from reference [21] has changed accordingly.
COMMENT #5: “In the 'Materials and Methods' section, please detail the materials used and the specific of in vivo experimental procedures, including the types of animals and tissues involved. Additionally, provide the approval number for the ethical review of the in vivo study.”
Response: We would like to clarify that we did not conduct any in vivo experiments as part of this study. Rather, comparisons to experimental data for in vivo simulations was done by comparing to the experimental data reported in:
[11] Hui, T. C. H.; Brace, C. L.; Hinshaw, J. L.; Quek, L. H. H.; Huang, I. K. H.; Kwan, J., et al. Microwave ablation of the liver in a live porcine model: The impact of power, time and total energy on ablation zone size and shape. Int.J. Hyperthermia 2020, 37(1), 668-676.
COMMENT #6: “Figure positioning should be carefully considered to avoid misleading reviewers. Typically, figures should correspond to the sequence of the 'Materials and Methods' section, with simulation results presented first, followed by experimental observations and quantification data. In this case, Figure 4 should be placed after the simulation results. Additionally, it is noted that the figure lacks a physical scale, which is essential for accurate interpretation of the data.”
Change in manuscript: We have placed Figure 4 (now Figure 9) at the end of the results section. We have renumbered all the figures accordingly. We have also included a scale in the new Figure 9.
COMMENT #7: “The paper includes the simulation study, yet the simulation results are presented solely in data form, omitting the visual simulation representation. Please specify.”
Response: We agree with the reviewer that a visual representation of some computational results would be interesting to show. For this reason, we have added two new figures (Figure 6 and Figure 8). Figure 6 shows the temperature profile and the thermal damage contour Ω=4.6 for the cases of 50, 75 and 100 W after 10 minutes of microwave application and under in vivo conditions. And Figure 8 shows the temperature profile and isoline Ω=4.6 at the end of all the protocols considered in Phase 2 under in vivo conditions.
Change in manuscript: We have added new Figures 6 and 8. The other figures have been renumbered accordingly. We have commented on the contents of Figures 6 and 8 and we relate them with other findings in the results section.
COMMENT #8: “In Figure 4, the tissue exhibits irregularity, and there is variability in the depth of electrode insertion. The quantified data, which includes ablation width (L) and depth (D), should be supplemented with a ratiometric analysis, such as the ablation area normalized to the electrode insertion depth. Furthermore, the text does not provide a precise description or explanation of the depicted phenomena in Figure 4.”
Response: Thank you for this important comment. During experimental procedures, we aimed to insert the applicator a minimum of 5 cm beneath the surface of the tissue. We did not keep track of the exact depth of insertion (beyond this minimum threshold), and it is difficult to estimate this accurately from the ablation images. For the water-cooled antennas employed in the present study, the ablation profile is not critically dependent on insertion depth beyond this ~5 cm threshold.
Change in manuscript: We have added a comment of Figure 4 (now Figure 9) to underline the most important facts that can be extracted from these pictures.
COMMENT #9: “Figure 5 is described in the text as 'computational model predictions'; however, it is labeled in the figure as 'ex vivo' and 'in vivo', which are experimental results.”
Response: The results of Figure 5 (now Figure 4) refer to computational simulations made using the characteristics of ex vivo and in vivo situations in order to mimic both scenarios. The label of the figures tries to distinguish both cases of computational results. As this may have been ambiguous, we have changed the caption of the figure to underline that these are computational results, not experimental. And the same in Figure 6 (now Figure 5).
Change in manuscript: We have changed the caption of Figures 5 and 6 (now Figures 4 and 5).
COMMENT #10: “The full text lacks a detailed statistical analysis to determine the significance of the observed differences.”
Response: Thank you for this suggestion because we think that determining the significance of the observed differences in ex vivo experiments of Phase 2 is a key point. For this reason, we have conducted a test with the values of D and L for the ex vivo experiments of Phase 2. Due to the amount of data and its statistical distribution, we have chosen the Kruskal-Wallis test as a rank-based nonparametric test that allows the comparison of more than two independent groups of data. Moreover, as a post-hoc comparison we used the Bonferrni procedure to make a pairwise comparisons between the average range of all protocols. We have used the software Statgraphics v18 to conduct the test with all the data obtained from the ex vivo experiments.
In Phase 1 we have also quantified some of the differences observed (we have specified terms like “negligible differences” or “small differences”).
Change in manuscript: We have included some sentences and comments in the paragraphs dealing with the results of Phase 1. And a paragraph in the results section to show all the results obtained from the Kruskal-Wallis test and the Bonferroni procedure for the ex vivo experiments of Phase 2.
COMMENT #11: “In Table 3, the sample number (n) should ideally range from 3 to 5; using only two data sets is not recommended.”
Response: We agree with the reviewer. We note that we had to exclude some experiments because the coagulation zone could not be reliably measured. This was because the presence of large vessels or the destruction of the tissue, did not allow us to properly measure the coagulation zone.
Change in manuscript: We have included a footnote in Table 3 to explain this.
COMMENT #12: “The presentation of data for Phase 1 and Phase 2 is inconsistent, which may lead to confusion. Therefore, it is recommended that a uniform diagrammatic representation be used for both phases to ensure clarity and comparability.”
Response: Thank you for this suggestion. We acknowledge the difference in style of presentation of the results in Phase 1 and Phase 2. In Phase 1, constant power protocols, one of the goals was to study how the ablation zone dimensions varied as a function of time and applied power level. As such, for the three applied power levels considered, in both experiments (in ex vivo tissue) and simulations, ablation zone dimensions were assessed at more than one time point. Experiments considered 5 min and 10 min ablations, whereas simulations afforded considerably more intermediate time points. In Phase 2, the goal was to assess ablation zones at the end of specific protocols as described in a prior study [11]. As such, ablation zone dimensions are only provided at one time point. Given these differences in goals of Phase 1 and Phase 2, and the associated differences in the data collected, we do not believe the data from both phases could be presented in a common diagrammatic manner.
Change in manuscript: In the revised manuscript, we have added two new figures illustrating thermal profiles at the end of ablations under in vivo conditions for both phase 1 (Figure 6) and figure (Figure 8).

Reviewer 2 Report
Comments and Suggestions for Authors
1. The power at the port boundary proposed in line 212 at 80%,different from the power at Phase 2 in line 215 at 65%. Why not take the same percentage?
2. Although it is mentioned in line 248 that the data in Figure 5 is a simulation result, it is recommended to include it in the caption as well, otherwise it may be mistaken as experimental data, and the same applies to Figure 6。
3. Line 295, Table 3 should be in the Phase 2.
4. In Phase 2, the simulation process of different power combinations were not explained.
5. It can be clearly seen from Figure 7 that the values of D and L are different in different power combinations, which does not match those data in Table 3,How to explain this?
Author Response
COMMENT #1: “The power at the port boundary proposed in line 212 at 80%, different from the power at Phase 2 in line 215 at 65%. Why not take the same percentage?”
Response: In Phase 1 we compared the computational results with our own ex vivo experiments, so we needed to reproduce the characteristics of our experimental microwave ablation system considering that the power loss in intervening cables was 20% of the specified generator power level. For this reason, the power applied at the port in the computational model was 80% of the specified generator power level.
In phase 2 the computational results were compared with ex vivo and in vivo experiments. The ex vivo experiments were conducted by our group, and thus, we had to use again 80% of the specified power level as a port condition in the computational model. However, we used the in vivo results reported in reference [11] to compare with the in vivo computational results. Thus, we had to consider the power loss related to [11] which was 35%, so the power applied at the port in the computational model was 65% of the specified power level at the generator reported in [11].
Change in manuscript: We agree with the reviewer that this needs to be clarified, so we have made changes in this paragraph trying to be more clear.
COMMENT #2: “Although it is mentioned in line 248 that the data in Figure 5 is a simulation result, it is recommended to include it in the caption as well, otherwise it may be mistaken as experimental data, and the same applies to Figure 6.”
Response: We agree with the reviewer that it is needed to underline that they refer to computational results.
Change in manuscript: We have changed the captions of Figures 5 and 6 (now Figures 4 and 5).
COMMENT #3: “Line 295, Table 3 should be in the Phase 2.”
Response: Thanks. We have changed it.
COMMENT #4: “In Phase 2, the simulation process of different power combinations were not explained.”
Response: Thank you for this comment. We have added more details about phase 2 simulations.
Change in manuscript: We have added the following paragraph to explain this at the end of section 2.1.5.: In phase 2, simulations considered ablations where the applied power level was varied as a function of time. For each of the considered protocols, applied power at the antenna input port was specified in a time-dependent manner, consistent with the protocols illustrated in Figure 2 and Figure 3. We note that the computational models for both phase 1 and phase 2 are coupled electromagnetic-bioheat transfer models, and as such, the time-harmonic electric field and electromagnetic power loss density in tissue were updated at each time step of the transient heat transfer solver. For phase 2 models, finer time steps were taken the solver consistent with the step changes at transitions.
COMMENT #5: “It can be clearly seen from Figure 7 that the values of D and L are different in different power combinations, which does not match those data in Table 3,How to explain this?”
Response: There are two main reasons that can explain this. The first is that the values of Table 3 are results obtained under ex vivo conditions, and the values of Figure 7 are obtained under in vivo conditions. The effect of the blood perfusion makes that the power protocol can have a different effect on the growth of the ablated zone. Secondly, the antenna and experimental setup used for the ex vivo and in vivo experiments were different. We developed the ex vivo experiments and we related in the manuscript the characteristics of the system. The in vivo experiments were developed by the group of Hui et al (reference [11]) using a different antenna. For the computer/theoretical modeling we have used always the same antenna as we mentioned in the manuscript (both ex vivo and in vivo simulations). For this reason, as we stated in the discussion, we could only make a qualitative comparison of the in vivo results in Phase 2.

Reviewer 3 Report
Comments and Suggestions for Authors
In this work, simulation studies and experimental results on liver microwave ablation have been compared aiming to provide information about the influence of energy delivery settings (power and time of ablation) on the ablated volumes. Moreover, in vivo previously performed studies in literature have been considered for comparative study. Modeling and experimental findings showed that the level of applied energy is the main cause of differences in the size of the ablation zone. Moreover, the existence of thresholds under ex vivo and in vivo conditions of continuous MWA show that long time and high-power applications do not always produce better results in terms of coagulation size and shape.
Few suggestions:
- Fig. 1: the resolution could be improved.
- Fig. 2: Labels for x- and y-axis are missing.
- In Fig. 4, the selected tissue section for each test could be highlighted for better understanding the criteria for identifying the ablation zones.
Author Response
COMMENT #1: “Fig. 1: the resolution could be improved.”
Response: Done.
COMMENT #2: “Fig. 2: Labels for x- and y-axis are missing.”
Response: We have added the labels.
COMMENT #3: “In Fig. 4, the selected tissue section for each test could be highlighted for better understanding the criteria for identifying the ablation zones.”
Response: Thank you for this important comment. In Figure 4 (now Figure 9), image in the top left panel, we have demarcated the extent of the ablation zone short axis (D) and long axis (L) diameters, as an illustration of how the ablation zone dimensions were recorded. We note that the reported dimensions were based on measurements on the physical tissue samples with a ruler. As such, we feel it may be a bit misleading to annotate the extents of the ablation zone on each image, as we do not intend to convey that the reported measurements were derived from the images.
Change in manuscript: We have added the following text to the Figure 9 caption: “Images of …with ex vivo experiments. As an example, the 50 W, 10 min image is annotated to illustrate how extents of the ablation zone short axis (D) and long axis (L) diameters were determined on tissue samples.”
